# Compensated Advanced Chronic Liver Disease and Steatosis in Patients with Type 2 Diabetes as Assessed through Shear Wave Measurements and Attenuation Measurements

**DOI:** 10.3390/biomedicines12020323

**Published:** 2024-01-30

**Authors:** Mislav Barisic-Jaman, Marko Milosevic, Viktoria Skurla, David Dohoczky, Josip Stojic, Petra Dinjar Kujundzic, Maja Cigrovski Berkovic, Ana Majic-Tengg, Ana Matijaca, Tomo Lucijanic, Mirjana Kardum-Pejic, Vlatka Pandzic Jaksic, Srecko Marusic, Ivica Grgurevic

**Affiliations:** 1Department of Gastroenterology, Hepatology and Clinical Nutrition, University Hospital Dubrava, 10000 Zagreb, Croatia; mislav.barisic.jaman@gmail.com (M.B.-J.); s.viktoria.94@gmail.com (V.S.); david.dohoczky@gmail.com (D.D.); josip.stojic95@gmail.com (J.S.); petra.dinjar@gmail.com (P.D.K.); ivicag72@gmail.com (I.G.); 2Department of Endocrinology, Diabetes, Diseases of Metabolism and Clinical Pharmacology, University Hospital Dubrava, 10000 Zagreb, Croatia; maja.cigrovskiberkovic@gmail.com (M.C.B.); ana.majic89@gmail.com (A.M.-T.); anamatijaca@gmail.com (A.M.); tomolucijanic@gmail.com (T.L.); mirjana.kardum@gmail.com (M.K.-P.); marusic.srecko@gmail.com (S.M.); 3Department of Sport and Exercise Medicine, Faculty of Kinesiology, University of Zagreb, 10000 Zagreb, Croatia; 4School of Medicine, University of Zagreb, 10000 Zagreb, Croatia; 5Faculty of Pharmacy and Biochemistry, University of Zagreb, 10000 Zagreb, Croatia

**Keywords:** MASLD, diabetes mellitus, ultrasound, elastography, liver fibrosis, liver steatosis

## Abstract

Patients with type 2 diabetes (T2D) are at risk of developing metabolic dysfunction-associated steatotic liver disease (MASLD). We investigated the prevalence of compensated advanced chronic liver disease (cACLD) and steatosis in patients with T2D using the new non-invasive diagnostic methods of shear wave measurements (SWMs) and attenuation (ATT) measurements in comparison with those of vibration-controlled transient elastography (VCTE) and the controlled attenuation parameter (CAP), which served as the reference methods. Among 214 T2D patients, steatosis at any grade and cACLD were revealed in 134 (62.6%) and 19 (8.9%) patients, respectively. SWMs showed a high correlation with VCTE (Spearman’s *ρ* = 0.641), whereas SWMs produced lower (mean of −0.7 kPa) liver stiffness measurements (LSMs) overall. At a LSM of >11.0 kPa (Youden), SWMs had an AUROC of 0.951 that was used to diagnose cACLD (defined as a LSM of >15 kPa through VCTE) with 84.2% sensitivity and 96.4% specificity. The performance of ATT measurements in diagnosing liver steatosis at any grade (defined as the CAP of ≥274 dB/m) was suboptimal (AUROC of 0.744 at the ATT measurement cut-off of >0.63 dB/cm/MHz (Youden) with 59% sensitivity and 81.2% specificity). In conclusion, the prevalence of liver steatosis and previously unrecognized cACLD in patients with T2D is high and SWMs appear to be a reliable diagnostic method for this purpose, whereas further investigation is needed to optimize the diagnostic performance of ATT measurements.

## 1. Introduction

The proportion of type 2 diabetes (T2D) has escalated over the past few decades, thereby making it a prevalent medical condition and a significant public health challenge of the twenty-first century [1]. In addition to its well-recognized complications affecting the cardiocirculatory system, kidneys, eyes, and peripheral nerves, it is frequently accompanied by metabolic dysfunction-associated steatotic liver disease (MASLD) [2]. MASLD is a major cause of liver disease worldwide, with a global prevalence of 32.4% in the general population, which is almost 60% in patients with T2D [3]. Metabolic comorbidities associated with MASLD include obesity, T2D, hyperlipidemia, hypertension, and metabolic syndrome [4]. By 2040, over half the adult population worldwide is forecasted to have MASLD [5]. With approximately 35% of MASLD patients eventually developing progressive forms of the disease resulting in nonalcoholic steatohepatitis (NASH), fibrosis, cirrhosis, and hepatocellular carcinoma (HCC), the substantial prevalence of this condition poses a significant burden to the healthcare system due to its impact on a large portion of the population [6,7,8]. Liver fibrosis is the most important histological feature associated with the adverse clinical outcomes for patients with MASLD [9]. Thus, it is of paramount importance to find reliable non-invasive test(s) (NIT(s)) capable of the early recognition of the presence of liver fibrosis, which is especially important for high-risk populations such as that of T2D patients who have a ten times higher prevalence of advanced fibrosis in comparison with the general population [10]. Recognizing the presence of liver steatosis early is also clinically relevant as it is the background of the development of MASLD [11]. Liver biopsy is the gold standard for diagnosing liver fibrosis, steatosis, and inflammation; however, it is associated with limitations such as sampling error, cost, and risk of complications [12]. Elastography is the most widely used among the non-invasive methods and vibration-controlled transient elastography (VCTE), which is used to assess the amount of liver fibrosis through liver stiffness measurements (LSMs), is the most validated form of it [13]. Additionally, it is possible to assess the presence and severity of liver steatosis using the same device through the controlled attenuation parameter (CAP) that measures the attenuation of the ultrasound beam. Although widely available, user-friendly, and supported by a high amount of scientific evidence, VCTE requires a special device—FibroScan (Echosens, Paris, France)—that cannot perform the conventional ultrasound, which is needed by patients with chronic liver disease to screen for HCC and other liver-related complications. It also has limitations such as inadequate measurements in severely obese patients and the inability to obtain reliable measurements in the presence of perihepatic ascites [14,15]. On the other hand, acoustic radiation force impulse (ARFI)-based techniques, as well as either point shear wave elastography (pSWE) or two-dimensional SWE, are integrated into regular ultrasound devices and thus might, in theory, overcome most of these limitations and offer the possibility of the comprehensive assessment of liver health during the same ultrasound investigation [16,17]. Shear wave measurements (SWMs, which are a representative of pSWE) and attenuation (ATT) measurements represent relatively new methods for the assessment of liver fibrosis and steatosis, with a limited amount of data being available to support their clinical usefulness [18]. Hence, we aimed to evaluate the diagnostic performance of SWMs and ATT measurements to assess patients with T2D for the prevalence of compensated advanced chronic liver disease (cACLD) as the adverse prognostic indicator and liver steatosis as the indicator of MASLD using VCTE as the reference method.

## 2. Materials and Methods

### 2.1. Patients

In this study, patients with T2D who attended the diabetes outpatient clinic from April 2022 to December 2023 were prospectively enrolled. Four patients per week were enrolled from two diabetes outpatient clinics, all of whom provided written consent to participate in the study. All patients were required to have routine bloodwork results that were not older than 1 month (including, at least, those for complete blood count, liver function tests, lipids and glucose, iron studies, creatinine, and urine analysis). Exclusion criteria included conditions that can affect the reliability of elastographic measurements such as liver congestion due to heart failure (characterized by dilated hepatic veins with a stagnant flow profile), high inflammatory activity in the liver (ALT > 5 times above the reference values), biliary obstruction (dilated intrahepatic bile ducts with elevated levels of GGT, ALP, and/or bilirubin), hemodialysis, liver tumors, and pregnancy. Patients were also excluded if they had other liver diseases (viral hepatitis B or C, excessive alcohol consumption (>30 g/day for men and >20 g/day for women), hemochromatosis (elevated ferritin levels, Fe/TIBC > 50%), Wilson’s disease, autoimmune liver diseases, autoimmune cholangiopathies, and drug-induced liver damage. A total of 243 patients agreed to participate in the study. Of them, 2 (0.8%) were excluded due to malignant liver tumors that were diagnosed through further imaging studies, and 4 (1.6%) were excluded due to the presence of other chronic liver diseases (2 due to primary biliary cholangitis, 1 due to chronic hepatitis B infection, and 1 due to hereditary hemochromatosis). The remaining 237 patients underwent further investigation, i.e., measurements of the CAP, ATT, and LSMs through SWMs and VCTE. The study flowchart is depicted in Figure 1.

### 2.2. Methods

Patients who met the inclusion criteria underwent a liver ultrasound, elastography and attenuation analysis, which were performed on the same day if they were fasting or within 2 weeks from the outpatient visit. The liver ultrasound was performed first, followed by shear wave measurements (SWMs) and attenuation (ATT) measurements as the investigational methods for the non-invasive assessment of liver fibrosis and steatosis. The examinations were performed using the Fujifilm Arietta 65 device (Fujifilm Healthcare, Tokyo, Japan) with a convex broadband probe operating at a frequency range of 1–5 MHz. During the same visit, liver stiffness measurements (LSMs) using vibration-controlled transient elastography (VCTE) and attenuation analysis using the controlled attenuation parameter (CAP) method were conducted. These served as reference methods for determining liver fibrosis and steatosis utilizing the FibroScan^®^ Compact 530 device (Echosens, Paris, France) equipped with the M or XL Probe. Use of the M or XL probe was based upon the suggestion of the probe selection tool, which is an automated software integrated into the FibroScan device. All elastographic and attenuation analyses were performed on fasting patients in a supine position with the right arm in the maximal abduction during a short apnea in the neutral breathing position. The ultrasound/FibroScan probe was placed in the intercostal space mostly in the midaxillary line over the right lobe of the liver, which was undertaken according to the international guidelines [13]. By using an ultrasound, the area without artifacts, large blood vessels, and biliary ducts was chosen for further interrogation. For SWMs, the region of interest was placed at least 1.5 cm below the liver capsule. For both methods, at least 10 measurements were taken per method, and the measurement was considered reliable if the interquartile range/median (IQR/M) was <0.3 [19]. As the reference values for LSMs using the VCTE method thresholds were employed in line with Baveno consensus [20]. Patients with a LSM of <5 kPa were considered to have no fibrosis, those with LSMs between 5 and 10 kPa were considered to have some fibroses but not advanced fibrosis (this was clinically used to rule out advanced fibrosis/cirrhosis), LSM values of 10–15 kPa were considered to be suggestive of cACLD, whereas LSMs of >15 kPa were considered to be highly suggestive of cACLD (a surrogate indicator of advanced fibrosis/cirrhosis) [20]. To assess the presence of any degree of steatosis (S ≥ 1), a CAP value of 274 kPa was used as the reference, which was based on the study by Eddowes PJ et al. that reported a sensitivity of >90% at this cut-off [21]. We did not test other cut-offs to further distinguish between the grades of steatosis as it was repeatedly shown that the performance of the CAP for this purpose was suboptimal [21,22]. According to a recent consensus document, MASLD is defined as the presence of liver steatosis in conjunction with at least one cardiometabolic risk factor and no other discernible cause [23]. Since T2D has pivotal cardiometabolic risk factors, we considered MASLD across all the T2D patients with a CAP of >274 dB/m, which suggests the presence of liver steatosis. The FIB 4 score was calculated using an on-line calculator (https://www.mdcalc.com/calc/2200/fibrosis-4-fib-4-index-liver-fibrosis; accessed on 10 January 2024), which was undertaken according to the formula FIB-4 = age (years) × AST [U/l]/(platelets [10^9^/L] × (ALT [U/L])^1/2^) from the original publication [24,25]. The APRI score also was calculated using an online calculator (https://www.mdcalc.com/calc/3094/ast-platelet-ratio-index-apri; accessed on 10 January 2024), which was undertaken according to the formula APRI = (AST in IU/L)/(AST upper limit of normal in IU/L)/(platelets in 10^9^/L) from the original publication [26].

### 2.3. Statistical Analysis

The Shapiro–Wilk test was utilized to assess the normality of the distribution of numerical variables. However, none of the numerical variables under analysis demonstrated a normal distribution. Instead, they were presented in terms of the median and interquartile range (IQR) and compared between groups using the Mann–Whitney U test. Categorical variables were represented as ratios and percentages and were compared between groups using the Χ^2^ test. Age was described using the median and range. Logistic regression was employed to examine the independent associations between various parameters. Only variables that were univariately significant were included in the model building process. The correlation between two numerical variables was tested using the Spearman correlation and is described using the Spearman coefficient of correlation *ρ*. The AUROC (area under the receiver operator characteristic curve) along with its corresponding 95% confidence interval (CI) was calculated to assess the ability of SWMs and ATT measurements in determining the fibrosis and steatosis stages as defined through VCTE. Optimal cut-offs for distinguishing between the presence of advanced fibrosis or steatosis were established using the Youden index [27], which aims to maximize the sum of sensitivity (sens.) and specificity (spec.). Furthermore, additional cut-offs were selected to maximize sensitivity and specificity for rule in and Rule out purposes. For each cut-off, sensitivity, specificity, the positive predictive value (PPV), the negative predictive value (NPV), the positive likelihood ratio (+LR), and the negative likelihood radio (-LR) were calculated. Statistical significance was defined as *p* values < 0.05. All statistical analyses were carried out using MedCalc statistical software version 22.016 (MedCalc Software Ltd., Ostend, Belgium).

### 2.4. Ethical Issues

This study was approved by the institutional ethics committee (No. 2022/0905-01), and all patients signed to provide their informed consent for participation in the study.

## 3. Results

Out of 237 patients, 23 (9,5%) were excluded from the study as they did not fulfill the reliability criterion of IQR/M ≤ 0.3 for either VCTE or SWMs. The final cohort comprised 214 patients with T2D, 112 (52.3%) males, an average age of 66 years, a median body mass index of 30.8 kg/m^2^, 74 (34.6%) overweight, 122 (57.5%) obese, 165 (78.6%) with arterial hypertension, 159 (75.7%) with dyslipidemia, and 49 (23%) smokers. As for VCTE, the M probe was used for 146 (68.5%) patients, the median LSM measured through VCTE was 5.9 kPa, and the median CAP was 291 dB/m). Seventy-three (34.1%) patients had LSMs that were <5 kPa through VCTE, and a further 108 (50.5%) patients had LSMs in the range of 5–10 kPa in whom cACLD can be ruled out based on Baveno recommendations. In 14 (6.5%) patients, LSMs were in a range between 10 and 15 kPa (suggestive of cACLD), whereas highly suggestive LSMs (>15 kPa) for cACLD were observed in 19 (8.9%) patients. We calculated the FIB-4 score for patients who had LSMs of ≥10 kPa (suggestive of cACLD, N = 33, based on VCTE results). Among them, 4 (12.1%), 15 (45.5%), and 14 (42.4%) patients had FIB-4 scores of <1.3, 1.3–2.67, and >2.67, respectively. The median SWM of the overall cohort of 214 patients was 5.5 kPa, and the median ATT measurement was 0.61 dB/cm/MHz (Table 1). The platelet count and prothrombin time decreased, while the aspartate aminotransferase (AST), alanine aminotransferase (ALT), high density lipoprotein (HDL), FIB-4 score, and APRI score increased with more advanced stages of liver disease (*p* < 0.05 for all analyses using the Mann–Whitney U test, Table 1). We observed a high correlation (Figure 2) between LSMs that were measured through VCTE and SWMs (Spearman’s *ρ* = 0.641, 95 CI: 0.554–0.713, *p* < 0.001)), whereas SWMs produced lower (mean of −0.7 unit, Bland–Altman) LSM values in comparison with VCTE (Figure 3). When analyzing patients with LSMs of < 10 kPa (through VCTE), SWMs produced slightly lower but very close values in comparison with VCTE (mean of – 0.1 unit, Bland–Altman). In patients with higher LSM values measured through VCTE (LSM ≥ 10 kPa), SWMs produced significantly lower values in comparison with VCTE (mean of −4.2 units, Bland–Altman) (Figure 3).

At the LSM cut-off 5.3 kPa (Youden), SWMs had an AUROC of 0.798 to differentiate the patients according to the presence/absence of any fibrosis (defined through the VCTE LSM cut-off of 5 kPa) with a sensitivity of 72.3%, a specificity of 75.3%, a PPV of 85%, a NPV of 58.4%, a +LR of 2.93, and a −LR 0.37). LSM cut-offs through SWMs that were optimized (>90% sensitivity and specificity) for ruling in and ruling out any fibrosis (LSMs of </> 5 kPa through VCTE) were, respectively, ≥6.4 kPa (specif. 90.4%) and ≤4.4 kPa (sens. of 90.1%). The presence of any fibrosis would be missed in 29.7%, 56%, and 9.9% patients using reported the SWM cut-offs (>5.3, ≥6.4, and ≤4.4 kPa, respectively) (Table 2). At the LSM cut-off of 8.3 kPa (Youden), SWMs had an AUROC of 0.923 to differentiate between the patients without and those under suspicion of cACLD (defined using the VCTE LSM of 10 kPa), with a sensitivity of 84.8%, a specificity of 95%, a PPV 75.5%, a NPV 97.2%, a +LR 17.06, and a −LR 0.16) (Figure 4). The LSM cut-off through SWMs that was optimized (>90% sensitivity and specificity) for ruling out cACLD was ≤6.3 kPa (sens. of 90.9%), whereas the SWM value that could rule in the VCTE measurement of ≥10 kPa (suspicion of cACLD) was ≥9.9 kPa (specif. 97.2%). The presence of cACLD would be missed in 15.2%, 27.3%, and 9.1% patients using the reported SWM cut-offs (>8.3, ≥9.9, and ≤6.3 kPa, respectively) (Table 2). At the LSM cut-off of 11.0 kPa (Youden), SWMs had an AUROC of 0.951 to diagnose cACLD (defined as LSMs of ≥15 kPa through VCTE, highly suggestive) with a sensitivity of 84.2%, a specificity of 96.4%, a PPV 69.5%, a NPV 98.4%, a +LR 23.46, and a −LR 0.16. LSM cut-offs through SWMs that were optimized (>90% sensitivity and specificity) for ruling in and ruling out cACLD were, respectively, ≥15.1 kPa (specif. 99.5%) and ≤6.4 kPa (sens. of 94.7%). The presence of cACLD would be missed in 15.8%, 26.3%, and 5.3% patients using reported the SWM cut-offs (>11.0, ≥15.1, and ≤6.4 kPa, respectively) (Table 2). When analyzing the optimal SWM cut-offs in patients with confirmed MASLD (CAP ≥ 274 dB/m, N = 134 patients), we obtained similar results. SWM cut-offs (Youden) corresponding to the VCTE cut-off of 10 kPa were the same as those of the overall cohort (8.3 kPa, AUROC of 0.908, sens. of 85.7%, spec. of 90.2%) and >11.0 kPa for a VCTE measurement of > 15 kPa (highly suggestive of cACLD, AUROC 0.913, sens. of 80%, spec. of 92.6%). The optimized SWM cut-off for ruling out cACLD (LSM of < 10 kPa through VCTE) was ≤6.4 kPa (sens. of 90.5%, spec. of 74.1%), whereas the optimized cut-off for ruling in cACLD (LSMs of > 15 kPa through VCTE) was ≥16.9 kPa (sens. of 60%, spec. of 96.6%) (Table 3). In the multivariate analysis using logistic regression, only platelet count (OR 0.973, 95% CI 0.948–0.998, *p* = 0.04), AST (OR 1.195, 95% CI 1.026–1.392, *p* = 0.02), and LSMs that were measured through VCTE (OR 1.376, 95% CI 1.160–1.632, *p* < 0.001) were independently and significantly associated with the SWM value (8.3 kPa), thereby indicating a suspicion of cACLD (Appendix A). On the other hand, in multivariate analysis, which only used logistic regression LSMs that were measured through VCTE (OR 1.371, 95% CI 1.151–1.633, *p*< 0.001), was independently and significantly associated with the SWM value (11 kPa) that was highly suggestive of cACLD (Appendix A).

Liver steatosis at any grade (CAP ≥ 274 dB/m) was detected in 134 (62.6%) patients (Table 4). The correlation between the ATT measurement and the CAP was statistically significant but weak (Spearman’s *ρ* = 0.422, 95 CI: 0.305–0.526, *p* < 0.001) (Figure 5), and the overall diagnostic performance of ATT measurements in diagnosing liver steatosis at any grade (CAP ≥ 274 dB/m) was suboptimal (AUROC of 0.744 at the ATT measurement cut-off of > 0.63 dB/cm/MHz (Youden), with a sens. of of 59%, a specif. of 81.2%, a PPV 84%, a NPV 54.2%, a +LR 3.14, and a −LR 0.51). The ATT measurement cut-offs that were optimized for ruling in and ruling out any steatosis were ≥0.74 dB/cm/MHz (specif. 91.2%), and ≤0.49 dB/cm/MHz (sens. of 90.3%), respectively. The presence of steatosis would be missed in 41%, 65.7%, and 9.7% of cases using the reported ATT measurement cut-offs (Table 4). In a multivariate analysis using logistic regression LSMs that were measured through VCTE (OR 1.105, 95%CI 1.011–1.208, *p* = 0.03) and the CAP (OR 1.112, 95% CI 1.0002–1.018, *p* = 0.04) were independently and significantly associated with the ATT value indicating the presence of steatosis (Appendix A).

## 4. Discussion

In this study, we evaluated the performance of SWMs and ATT measurements as relatively new methods to non-invasively search for the presence of liver steatosis and cACLD in patients with T2D using VCTE as the reference method. Our findings highlight the potential of SWMs as a reliable tool for assessing the presence of cACLD as the indicator of advanced fibrosis in the population, whereas the performance of ATT measurements in detecting steatosis was suboptimal.

Our study is significant for two reasons. Firstly, we investigated the prevalence of liver steatosis and cACLD in the population of patients with T2D who are at high risk of developing and progressing to metabolic dysfunction-associated steatotic liver disease (MASLD). Specifically, MASLD was identified in 62% of the patients in our study, which is approximately consistent with data from other research [3]. It should be noted that the presence of T2D accelerates the progression of MASLD to cirrhosis and hepatocellular carcinoma (HCC). Additionally, patients with T2D who also have MASLD have a >2× higher risk of overall mortality compared with patients without MASLD [4,28,29,30,31]. Although awareness of the need for screening high-risk populations for unrecognized liver disease is increasing, it is still not systematically implemented in routine practice for patients with T2D [11,32,33]. According to our results, 8.9% of patients had a high risk of cACLD, significantly increasing their risk of overall mortality and the risk of poor liver disease outcomes alongside metabolic comorbidities. LSMs of ≥10 kPa through VCTE that were traditionally used to delineate the presence of advanced fibrosis do not seem not accurate as only 45% of this group of patients had the FIB-4 score of >2.67 as the biochemical indicator of advanced fibrosis. This is not surprising as it has been previously demonstrated that VCTE overestimates the fibrosis stage in around 50% of patients with T2D, who were assumed to have advanced fibrosis based on LSMs [34]. Therefore LSMs of >15 kPa, as suggested by the Baveno consensus, might be more appropriate in this context. These things considered, almost 9% of patients with T2D had advanced chronic liver disease that was previously unrecognized, and thus it is of great importance to have reliable methods available for the early detection of liver steatosis and fibrosis in these patients to prevent undesirable outcomes in a timely manner. Although numerous non-invasive methods are available in clinical practice, including both biochemical and imaging-based methods such as elastography and attenuation analysis, each new method enhances accessibility, thereby improving the coverage of at-risk patients for liver disease development. Of note, whereas VCTE has been proven superior in stratifying the fibrosis risk and the presence of cACLD but might not be reliably measured in up to 20% of patients, especially in those who are obese, biochemical methods have almost universal applicability as they only rely on obtaining a blood sample [35,36].

In this regard, the second reason we consider this study to be significant lies in the fact that we evaluated the diagnostic performance of SWMs and ATT measurements as relatively new methods for diagnosing liver fibrosis and steatosis in this specific group of patients. Only a few studies have thus far analyzed these two methods for their performances in diagnosing liver steatosis and fibrosis but not specifically in patients with T2D. In a study conducted on a cohort of 445 patients with chronic hepatitis C that used VCTE as the reference method as well, the best performing SWM cut-off for ruling in significant fibrosis was 6.78 kPa (AUROC of 0.92, sens. of 76.9%, spec. of 90.3%), and for ruling in liver cirrhosis the best cut-off was 9.15 kPa (AUROC 0.94, sens. of 83.3%, spec. of 90.1%) [37]. The VCTE cut-offs that were used in the mentioned study as the reference were 7 kPa for significant fibrosis (F ≥ 2) and 12 kPa for cirrhosis (F = 4). In a Japanese study, the diagnostic performance of ATT measurements was examined for a cohort of 351 patients with a mixed etiology of chronic liver disease (mostly hepatitis C, 52%) in comparison with a liver biopsy that served as the reference for grading liver steatosis. The best performing ATT measurement cut-off for steatosis at any grade (≥S1) was 0.62 dB/cm/MHz (AUROC of 0.79, with 72% sensitivity and 72% specificity), which is in accordance with our results [38]. As an illustration of the diagnostic performance of the elastography methods other than pSWE, a study evaluated the performance of two-dimensional shear wave elastography (2D-SWE) in 552 patients with MASLD and compared it with VCTE, which was the reference method, thus revealing a strong correlation between 2D-SWE and VCTE (Pearson’s R= 0.84 for diagnosing severe fibrosis and Pearson’ R = 0.658 for diagnosing cirrhosis). The proportion of patients with at least severe fibrosis was 15.9%, which is very similar to our results, and the respective LSM cut-offs and AUROCs through 2DSWE for severe fibrosis (8.9 kPa, AUROC 0.988) and cirrhosis (11 kPa, AUROC of 0.998) were similar to our results as well [39]. We demonstrated a very high correlation between the LSM values measured through the SWM method and the reference method, VCTE, that was used in our study. In a multivariate analysis using logistic regression, we found that only the values of platelets (Plt), aspartate aminotransferase (AST), and LSMs measured through the VCTE method significantly influenced the SWM result indicating suspicion of cACLD (Appendix A). This is expected when lower Plt values and higher AST values indicate a higher stage of fibrosis, i.e., advanced liver disease. The threshold values for distinguishing patients with/without cACLD at 8.3 kPa are slightly lower compared with VCTE using FibroScan. The same holds true when considering the SWM values that would be highly suggestive of cACLD, and this value was set at 11 kPa in comparison with the value of 15 kPa, which was suggested by the Baveno consensus for VCTE. This is in line with previous observations that methods based on ARFI (acoustic radiation force impulse) technology follow the rule of four. Thus, according to the recommendations of the society of radiologists in ultrasound, the corresponding SWM values for VCTE of 10 and 15 kPa would be around 9 kPa and 13 kPa, respectively [16]. In our cohort, SWMs produced LSM values that were 0.7 and 4.2 kPa lower on average among the patients with a VCTE measurement of <10 kPa and >10 kPa, respectively, and this phenomenon has also been described for other point shear wave elastography methods [40]. Thus, it is possible that the lower LSM values obtained through SWMs in our study are connected to the technical solution utilized in the ultrasound vendor that was used but might also have been influenced by the composition of the investigated cohort, where many participants had very low LSM values, thus leading to the spectrum bias. However, the values optimized for detecting cACLD (with >90% specificity) are very similar to the VCTE values and amount to >15.1 kPa and >16.9kPa using the SWM method across the overall T2D cohort and patients with verified MASLD, respectively. Of course, by dichotomizing LSMs according to the optimal values for excluding and determining a specific stage of fibrosis, certain patients would remain in the “gray zone” between these threshold values. This has also been observed in studies evaluating other elastographic methods, and the use of an additional non-invasive test is recommended in such cases for more accurate risk stratification. In contrast, ATT measurements showed a poor correlation with the CAP and exhibited a suboptimal diagnostic performance for detecting liver steatosis at any grade (AUROC of 0.744 using the reference CAP cut-off of 274 dB/m). Using the reported ATT measurement cut-off would result in a substantial number of cases of liver steatosis being missed. In a multivariate analysis using logistic regression, we found that only the CAP value and LSMs measured through VCTE significantly influenced the ATT measurement result (Appendix A). Whereas it is intuitive to expect the influence of the CAP on ATT measurement results as they both measure steatosis, it is much more difficult to speculate on the reasons for the observed influence of LSMs through VCTE on the ATT measurement results as a correlation was not observed vice versa. Overall, the suboptimal performance of the ATT measurements may be a consequence of using the CAP as the reference method, which is not entirely optimal for assessing steatosis. CAP has a relatively acceptable performance for detecting any degree of steatosis (S ≥ 1), but it is not reliable enough for further differentiation between individual steatosis stages [21,22]. In this regard, we conclude that it would be necessary to evaluate this new ATT measurement method against another more reliable reference method for assessing liver steatosis, such as liver biopsy or magnetic resonance imaging (MRI). Nevertheless, the threshold values for an ATT measurement of 0.63 dB/cm/MHz for the presence of any degree of steatosis obtained in our study are very similar to the values obtained in other studies that used the same or similar non-invasive technology, thereby making them promising [38,41].

Our study has its limitations, primarily the fact that an optimal reference method for assessing liver steatosis and fibrosis, such as biopsy or MRI, was not used. Additionally, the total number of participants is relatively small, with there being potential spectrum bias of cACLD as the surrogate for advanced fibrosis distribution that could affect the obtained results. On the other hand, the study was conducted on a well-defined cohort of patients at a significant risk of having MASLD, thus representing a group of patients for whom research and the application of non-invasive methods for screening for unrecognized liver disease should be directed.

In conclusion, our study revealed a high prevalence of liver steatosis and cACLD among the patients with T2D and demonstrated SWMs as a reliable method for diagnosing cACLD in this population. The strong correlation with VCTE and the optimized cut-offs for ruling in and ruling out cACLD provide valuable guidance for clinical decision making. However, ATT measurements showed limited utility in diagnosing liver steatosis, thus highlighting the need for further investigation and comparing this method with already established reference standards such as liver biopsy or magnetic resonance.

## Figures and Tables

**Figure 1 biomedicines-12-00323-f001:**
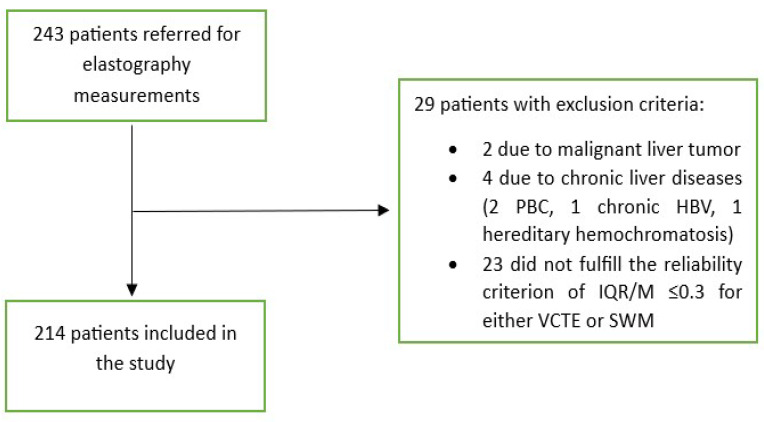
Flowchart of patient selection. Abbreviations as follows: HBV: hepatitis B virus; IQR: interquartile range; M: median; PBC: primary biliary cholangitis; SWM: shear wave measurement; VCTE: vibration-controlled transient elastography.

**Figure 2 biomedicines-12-00323-f002:**
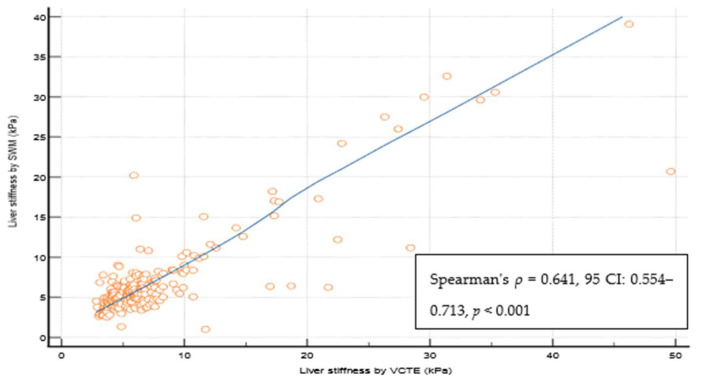
Spearman coefficient of the correlation of reliable liver stiffness measurements through VCTE and SWM. Abbreviations: CI: confidence interval; kPa: kilopascals; SWM: shear wave measurement; VCTE: vibration-controlled transient elastography.

**Figure 3 biomedicines-12-00323-f003:**
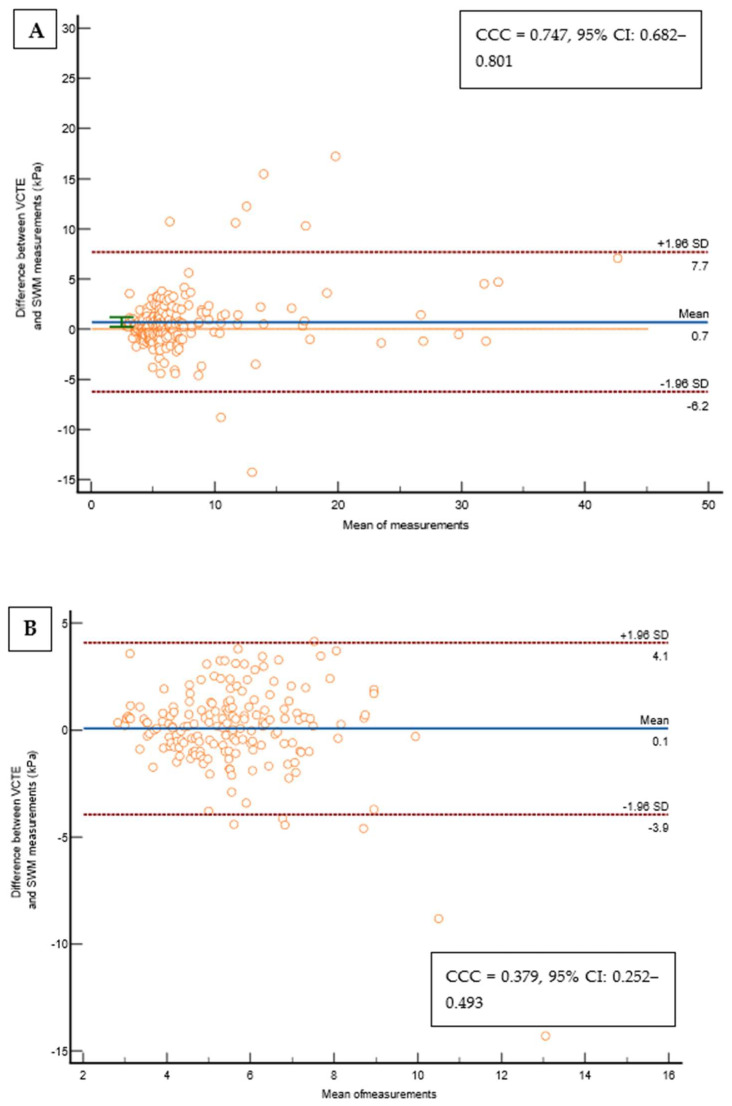
(**A**) Bland–Altman Leh plot comparing the difference between VCTE measurements and the SWMs of the overall cohort (N = 214) for a given mean of a measurement with lines of average difference and two 95% CI lines, with the Lin’s concordance coefficient shown on a pseudo-log10 scale. (**B**) Bland–Altman Leh plot comparing the difference between VCTE measurements and SWMs in patients with a VCTE measurement of <10 kPa for a given mean of a measurement with lines of average difference and two 95% CI lines, with the Lin’s concordance coefficient shown on a pseudo-log10 scale. (**C**) Bland–Altman Leh plot comparing the difference between VCTE measurements and SWMs in patients with a VCTE measurement of ≥10 kPa for a given mean of a measurement with lines of average difference and two 95% CI lines, with the Lin’s concordance coefficient shown on a pseudo-log10 scale. Abbreviations: CCC: concordance correlation coefficient; CI: confidence interval; kPa: kilopascals; SD: standard deviation; SWM: shear wave measurement; VCTE: vibration-controlled transient elastography.

**Figure 4 biomedicines-12-00323-f004:**
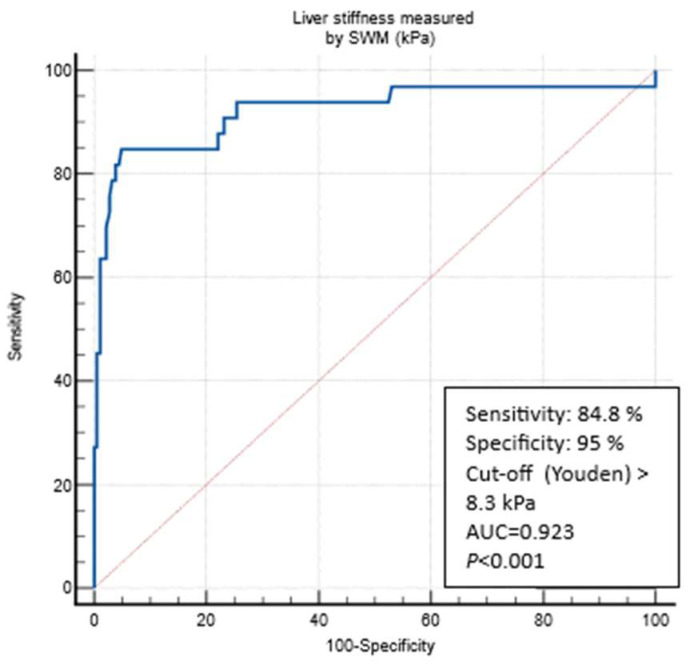
ROC curve analysis for predicting a VCTE measurement of ≥10 kPa through SWMs. Abbreviations: AUC: area under the curve; kPa: kilopascals; SWM: shear wave measurement; VCTE: vibration-controlled transient elastography.

**Figure 5 biomedicines-12-00323-f005:**
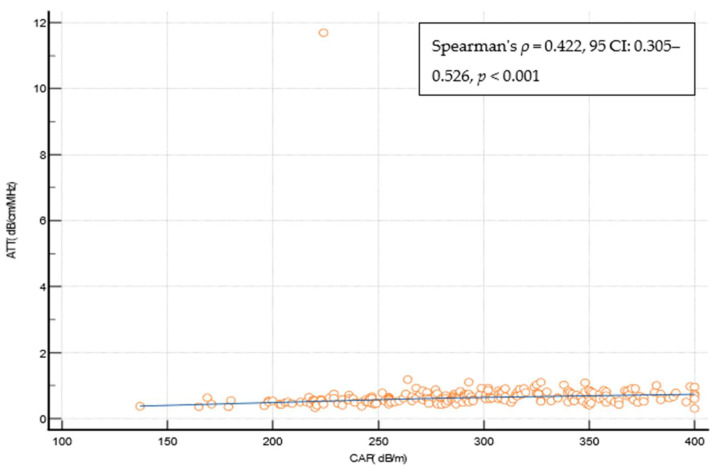
Spearman’s coefficient of correlation between the CAP and the ATT measurement. Abbreviations: ATT: attenuation measurement; CAP: controlled attenuation parameter; CI: confidence interval; dB/cm/MHz: decibel per centimeter per megahertz; dB/m: decibel per meter.

**Table 1 biomedicines-12-00323-t001:** Characteristics of the patients included in this study. *p*-values were calculated using the Kruskal–Wallis test and chi-squared test comparing the values of patients without cACLD (VCTE < 10 kPa), patients suggestive of cACLD (VCTE 10–15 kPa), and patients highly suggestive of cACLD (VCTE > 15 kPa). Significant *p*-values are in bold. Abbreviations ALT: alanine aminotransferase; AST: aspartate aminotransferase; ALP: alkaline phosphatase; APRI: AST to platelet ratio index; ATT: attenuation coefficient measurement; CAP: continuous attenuation parameter; CRP: C-reactive protein; dB/cm/MHz: decibel per centimeter per megahertz; dB/m: decibels per meter; FIB-4: fibrosis-4; GGT: gamma glutamyl transferase; g/L: grams per liter; HbA1c: glycated haemoglobin; HDL: high-density lipoprotein; IQR: interquartile range; kPa: kilopascal; LDL: low-density lipoprotein; M: median; µmol/L: micromoles per liter; mmol/L: millimoles per liter; N: number; MASLD: metabolic dysfunction-associated steatotic liver disease; PT: prothrombin time; SCD: skin to capsule distance; SWM: shear wave measurement; U/L: units per liter; VCTE: vibration-controlled transient elastography; Vs: shear wave speed.

	All Patients N = 214 (100%)	VCTE < 10 kPa N= 181 (84.6%)	VCTE 10–15 kPa N= 14 (6.5%)	VCTE > 15 kPa N= 19 (8.9%)	*p* Value
Age, years, median [IQR]	66 [56–71]	66 [57.8–71]	63 [57–66]	67 [56–71.5]	0.42
Sex 1—male 2—female	112 (52.3%) 102 (47.7%)	91 (50.3%) 90 (49.7%)	12 (85.7%) 2 (14.3%)	9 (47.4%) 10 (52.6%)	0.03
BMI, kg/m ^2^, median [IQR]	30.8 [27.8–35.4]	30.6 [27.8–35.9]	33.1 [28.7–34.9]	29.9 [26.6–32]	0.45
Obesity (BMI > 30 kg/m ^2^) Yes No	122 (57.5%) 90 (42.5%)	103 (57.2%) 77 (42.8%)	10 (71.4%) 4 (28.6%)	9 (50%) 9 (50%)	0.46
Arterial hypertension Yes No	165 (78.6%) 45 (21.4%)	140 (79.1%) 37 (20.9%)	8 (57.1%) 6 (42.9%)	17 (89.5%) 2 (10.5%)	0.07
Hyperlipidemia Yes No	159 (75.7%) 51 (24.3%)	133 (75.1%) 44 (24.9%)	12 (85.7%) 2 (14.3%)	14 (73.7%) 5 (26.2%)	0.66
Probe used M XL	146 (68.5%) 67 (31.5%)	122 (67.8%) 58 (32.2%)	9 (64.3% 5 (35.7%)	15 (78.9%) 4 (21.1%)	0.57
Smoking Yes No	49 (23%) 164 (77%)	38 (21.1%) 142 (78.9%)	5 (35.7%) 9 (64.3%)	6 (31.6%) 13 (68.4%)	0.29
Hematocrit, median [IQR]	0.426 [0.404–0.455]	0.425 [0.404–0.452]	0.464 [0.399–0.478]	0.435 [0.402–0.442]	0.29
Red cell count, G/L, median [IQR]	4.8 [4.5–5.1]	4.8 [4.5–5.1]	4.7 [4.6–5.2]	4.6 [4.3–4.9]	0.09
Platelets, G/L, median [IQR]	230 [196–276]	237 [207–279]	204 [180–229]	167 [142–193]	<0.001
PT (%), median [IQR]	102 [87–109]	103 [94–110]	102 [86–111]	87 [83–100]	0.03
Glucose, mmol/L, median [IQR]	7.8 [6.8–9.5]	7.8 [6.9–9.4]	7 [6.1–8]	8.3 [7.4–11.1]	0.06
HbA1c (%), median [IQR]	6.9 [6.3–7.8]	6.9 [6.3–7.9]	6.1 [5.9–6.9]	7.1 [6.6–7.8]	0.04
Creatinine, µmol/L, median [IQR]	78 [66–89]	79 [67–92]	75 [65–85]	72 [63–82]	0.17
AST, U/L, median [IQR]	23 [19–31]	22 [18–29]	34 [23–44]	49 [35–58]	<0.001
ALT, U/L, median [IQR]	25 [20–41]	24 [20–36]	31 [24–58]	53 [33–82]	<0.001
GGT, U/L, median [IQR]	29 [19–65]	28 [18–49]	82 [31–132]	65 [48–93]	<0.001
ALP, U/L, median [IQR]	69 [56–87]	68 [56–83]	78 [60–90]	83 [59–96]	0.27
Total cholesterol, mmol/L, median [IQR]	4.4 [3.7–5.4]	4.4 [3.7–5.4]	5 [3.9–5.3]	5 [3.8–5.7]	0.27
Triglycerides, mmol/L, median [IQR]	1.7 [1.2–2.4]	1.7 [1.2–2.5]	1.3 [1.1–1.6]	1.5 [1.1–1.9]	0.04
HDL, mmol/L, median [IQR]	1.2 [1–1.5]	1.2 [1–1.4]	1.3 [1.1–1.5]	1.5 [1.3–1.7]	0.02
LDL, mmol/L, median [IQR]	2.3 [1.8–3.1]	2.2 [1.8–2.9]	2.4 [2–3.6]	2.7 [1.8–3.7]	0.3
Albumins, g/L, median [IQR]	43 [41–46]	43 [41–46]	41 [38–43]	44 [42–46]	0.15
CRP, mg/L, mmol/L, median [IQR]	2.4 [1.3–4.5]	2.2 [1.1–4.6]	3.5 [2.7–4.4]	3.2 [1.9–4.3]	0.22
NAFLD fibrosis score, points, median [IQR]	−0.214 [−0.258–0.029]	−0.175 [−0.258–0.03]	−0.219 [−0.246–0.03]	−0.238 [−0.263–0.02]	0.53
FIB-4, points, median [IQR]	1.26 [0.91–1.65]	1.14 [0.87–1.55]	1.58 [1.37–2.21]	2.69 [1.61–3.74]	<0.001
APRI, points, median [IQR]	0.25 [0.191–0.396]	0.221 [0.183–0.32]	0.409 [0.323–0.603]	0.723 [0.55–0.861]	<0.001
VCTE, kPa, median [IQR]	5.9 [4.5–7.6]	5.6 [4.3–6.5]	11.4 [10.7–12.1]	22.8 [17.9–30.9]	<0.001
VCTE IQR/median, %, median [IQR]	15 [10–20]	14 [10–20]	18 [10–24]	17 [10–19]	0.80
SCD, cm, median [IQR]	1.9 [1.7–2.4]	1.9 [1.6–2.4]	2.3 [1.8–2.6]	1.9 [1.7–2.2]	0.26
CAP, dB/m, median [IQR]	291 [253–341]	291 [255–342]	295 [274–340]	278 [230–340]	0.84
SWM, kPa, median [IQR]	5.5 [4.5–7.1]	5.2 [4.4–6.3]	10.1 [8.4–11.6]	18.2 [12.9–29.1]	<0.001
SWM-Vs, m/s, median [IQR]	1.34 [1.21–1.5]	1.32 [1.2–1.44]	1.67 [1.51–1.8]	1.59 [1.21–2.53]	<0.001
ATT, dB/cm/MHz, median [IQR]	0.61 [0.53–0.74]	0.60 [0.52–0.72]	0.73 [0.6–0.85]	0.74 [0.45–0.81]	0.03

**Table 2 biomedicines-12-00323-t002:** Diagnostic performance of SWMs for different stages of liver diseases as defined through VCTE according to the Baveno recommendations (LSMs 5–9 kPa through VCTE for any level of fibrosis with cACLD excluded, LSMs 10–15 kPa were suggestive of cACLD, LSMs of >15 kPa were highly suggestive of cACLD through VCTE) in the overall cohort of patients with T2D (N = 214). Abbreviations: AUROC: area under the receiver operator characteristics; cACLD: compensated advanced chronic liver disease; kPa: kilopascals; −PV: negative likelihood ratio; NPV: negative predictive value; +LR: positive likelihood radio; PPV: positive predictive value; SWM: shear wave measurement; T2D: type 2 diabetes mellitus, VCTE: vibration-controlled transient elastography.

Stage	Cut-Off	AUROC	SWM Cut-Off, Stiffness (kPa)	Sensitivity, %	Specificity, %	+LR	−LR	PPV,%	NPV, %	Missed Cases
VCTE 5–9 kPa (cACLD excluded)	Youden	0.798 (95%CI: 0.738–0.850)	>5.3	72.3	75.3	2.93	0.37	85	58.4	42/141 (29.7%)
	Rule in		≥6.4	44	90.4	4.59	0.62	89.9	45.5	79/141 (56%)
	Rule out		≤4.4	90.1	43.8	1.6	0.23	75.6	69.6	14/141 (9.9%)
VCTE 10–15 kPa (suggestive of cACLD)	Youden	0.923 (95%CI: 0.879–0.955)	>8.3	84.8	95	17.06	0.16	75.5	97.2	5/33 (15.2%)
	Rule in		≥9.9	72.7	97.2	26.33	0.28	82.5	95.1	9/33 (27.3%)
	Rule out		≤6.3	90.9	76.8	3.92	0.12	41.6	97.9	3/33 (9.1%)
VCTE > 15 kPa (highly suggestive of cACLD	Youden	0.951 (95%CI: 0.913–0.976)	>11.0	84.2	96.4	23.46	0.16	69.5	98.4	3/19 (15.8%)
	Rule in		≥15.1	73.7	99.5	143.6	0.26	93.5	97.5	5/19 (26.3%)
	Rule out		≤6.4	94.7	70.3	3.42	0.07	22.8	99.3	1/19 (5.3%)

**Table 3 biomedicines-12-00323-t003:** Diagnostic performance of SWMs for different stages of liver diseases as defined through VCTE according to the Baveno recommendations (LSMs of 5–9 kPa through VCTE for any level of fibrosis with cACLD being excluded, LSMs of 10–15 kPa were suggestive of cACLD, LSMs of >15 kPa were highly suggestive of cACLD through VCTE) in patients with confirmed MASLD (N = 134). Abbreviations: AUROC: area under the receiver operator characteristics; cACLD: compensated advanced chronic liver disease; kPa: kilopascals; −PV: negative likelihood ratio; MASLD: metabolic dysfunction-associated steatotic liver disease; NPV: negative predictive value; +LR: positive likelihood radio; PPV: positive predictive value; SWM: shear wave measurement; VCTE: vibration-controlled transient elastography.

Stage	Cut-Off	AUROC	SWM Cut-Off, Stiffness (kPa)	Sensitivity, %	Specificity, %	+LR	−LR	PPV,%	NPV, %	Missed Cases
VCTE 5–9 kPa (cACLD excluded)	Youden	0.706 (95%CI: 0.641–0.767)	>4.6	90.1	44.2	1.62	0.22	59.1	83.3	10/101 (9.9%)
	Rule in		≥9	21.8	92.9	3.08	0.84	73.3	57.1	79/101 (78.2%)
	Rule out		≤4.5	91.1	42.5	1.58	0.21	58.6	84.2	9/101 (8.9%)
VCTE 10–15 kPa (suggestive of cACLD)	Youden	0.908 (95%CI: 0.861–0.943)	>8.3	85.7	90.2	8.71	0.16	48.7	98.3	3/21 (14.3%)
	Rule in		≥10.1	71.4	93.8	11.49	0.3	55.6	96.8	6/21 (28.6%)
	Rule out		≤6.4	90.5	74.1	3.49	0.13	27.5	98.6	2/21 (9.5%)
VCTE > 15 kPa (highly suggestive of cACLD	Youden	0.913 (95%CI: 0.866–0.947)	>11.0	80	92.6	10.88	0.22	34.8	98.9	2/10 (20%)
	Rule in		≥16.9	60	96.6	17.49	0.41	46.5	98	4/10 (40%)
	Rule out		≤ 6.4	90	70.6	3.06	0.14	13.1	99.3	1/10 (10%)

**Table 4 biomedicines-12-00323-t004:** Diagnostic performance of ATT measurements in detecting any grade (CAP > 274 dB/m) of liver steatosis. Abbreviations: ATT: attenuation measurement; AUROC: area under the receiver operator characteristics; CAP: controlled attenuation parameter; dB/m: decibel per meter; dB/cm/MHz: decibel per centimeter per megahertz; −LR: negative likelihood ratio; NPV: negative predictive value; +LR: positive likelihood radio; PPV: positive predictive value; S1: any stage of steatosis; sens.: sensitivity; spec.: specificity.

Stage of Steatosis	Cut-Off	AUROC	ATT Cut-Off, (dB/cm/MHz)	Sens, %	Spec., %	+LR	−LR	PPV, %	NPV, %	Missed Cases
≥S1 (CAP > 274 dB/m)	Youden	0.744 (95% CI 0.680–0.801)	> 0.63	59	81.2	3.14	0.51	84	54.2	42/141 (29.7%)
	Rule in		≥ 0.74	34.3	91.2	3.92	0.72	86.7	45.3	79/141 (56%)
	Rule out		≤ 0.49	90.3	31.2	1.31	0.31	68.7	65.8	14/141 (9.9%)

## Data Availability

The datasets used and/or analyzed during the current study are available from the corresponding author upon reasonable request.

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
