# Peer review of "Compensated Advanced Chronic Liver Disease and Steatosis in Patients with Type 2 Diabetes as Assessed through Shear Wave Measurements and Attenuation Measurements"

_biomedicines, 2024, doi:10.3390/biomedicines12020323_

Round 1

Reviewer 1 Report

Comments and Suggestions for Authors

The study is well organized, and the results are precise. It has significant limitations, such as the small number of participants and the absence of a liver biopsy as a reference. Nevertheless, the authors mention that in the discussion.

A) The only comment I have to make is that the authors should further discuss their results in comparison to those from other studies that had already compared VCTE to shear wave elastography (i.e. doi:10.1097/MEG.0000000000002412. Epub 2022).

B) Furthermore,  the authors should explain what is, in their opinion, the reason for the difference found between the results of VCTE and those of SWE.   

Comments on the Quality of English Language

Moderate English language editing is necessary

Reviewer 2 Report

Comments and Suggestions for Authors

The study by Barisic-Jaman M. et al. is a prospective study evaluating 214 patients with type 2 diabetes (T2D) with non-invasive diagnostic methods of Shear Wave Measurement (SWM) and Attenuation (ATT) Measurement in comparison to vibration-controlled transient elastography (VCTE) and controlled attenuation parameter (CAP) serving as the reference methods. The authors used the LSM cut-off of <5/5-10/>10 kPa to exclude fibrosis/identify some fibrosis/ identify advanced fibrosis. The authors concluded that the prevalence of liver steatosis and advanced fibrosis in patients with T2D is high, and SWM appears reliable diagnostic method, whereas further investigation is needed to optimize the diagnostic performance of ATT.

However, the current version has significant limitations and cannot be accepted in biomedicines

General comments

Studies evaluating the liver fibrosis stage in liver biopsies using the Baveno VI or VII criteria are lacking. The recent Consensus among patients with compensated cirrhosis or cACLD defined at the Baveno VI conference and Baveno VII the Criteria to identify cACLD (nor fibrosis stage) such as LSM values by transient elastography (TE) <10 kPa in the absence of other known clinical/imaging signs rule out cACLD; values between 10 and 15 kPa are suggestive of cACLD; and values >15 kPa are highly suggestive of cACLD (h􀄴ps://doi.org/10.1016/j.jhep.2021.12.022).

Therefore, authors should be more prudent in interpreting their results to avoid wrong conclusions. Please rewrite the title, abstract, and all the study eliminating "liver fibrosis" and using the cut-offs ”to “exclude”, “suggest” or “highly suggest” cACLD. 

Major comments

1.     Title. Authors should change “fibrosis” to “compensated advanced chronic liver disease (cACLD)”

2.     Abstract. Please include the cut-off definitions of cACLD and rewrite the results and conclusions. Avoid wrong conclusions.

3.     Introduction. Please, include appropriate references for lines 48-51 and lines 71-73

4.     Patients. Please, include the patients without fulfilling reliability criteria such as exclusion criteria

5.     Methods. Please include descriptions and references for APRI and FIB-4. Please include the cut-off definitions of cACLD. Please include the definition of MASLD. 

6.     Results and Tables. Please categorize the results according to patients with MASLD and without MASLD and those without cACLD and with. 

7.     Table 1. Please include the “interquartile range (IQR)” as a range

8.     Figure 2. Please show the SWM in kPa without a logarithmic scale

9.     Figure 3. Please show the dispersion of SWM after categorizing the patients in those without cACLD (< 10 kPa) and with cACLD (>10 kPa)

10.  Figure 4. Please include the best cut-off in the AUROC

11.  Figure 5. Please show the attenuation of SWM without a logarithmic scale

12.  Discussion. Please focus the discussion on cACLD and not on fibrosis. Please, include a paragraph regarding the lower applicability of TE and SWE compared to APRI and FIB-4 due to the excluded patients (almost 10%) not fulfilling the reliability criterion with elastography. Please discuss that higher VCTE values can overestimate ATT values. 

13.  Conclusion. Please abbreviate the conclusion section and avoid wrong conclusions

Round 2

Reviewer 1 Report

Comments and Suggestions for Authors

The revised version of the manuscript is absolutely better than the first version

All of my comments have appropriately been answered 

I have nothing to add

Reviewer 2 Report

Comments and Suggestions for Authors

The authors have made modifications to the original manuscript based on the reviewers’ comments and advice improving the quality of their study. 

Now, the manuscript is suitable for publication in biomedicines